# Diselenide-bond replacement of the external disulfide bond of insulin increases its oligomerization leading to sustained activity

Kenta Arai [1,2,14✉], Masaki Okumura [3,14], Young-Ho Lee[4,5,6,7,14], Hidekazu Katayama [8], Kenji Mizutani[9], Yuxi Lin [4], Sam-Yong Park [9], Kaichiro Sawada [10], Masao Toyoda [10], Hironobu Hojo[11], Kenji Inaba [12,13] & Michio Iwaoka [1,2✉]

Seleno-insulin, a class of artificial insulin analogs, in which one of the three disulfide-bonds (S-S's) of wild-type insulin (Ins) is replaced by a diselenide-bond (Se-Se), is attracting attention for its unique chemical and physiological properties that differ from those of Ins. Previously, we pioneered the development of a [C7U$^A$,C7U$^B$] analog of bovine pancreatic insulin (SeIns) as the first example, and demonstrated its high resistance against insulin-degrading enzyme (IDE). In this study, the conditions for the synthesis of SeIns via native chain assembly (NCA) were optimized to attain a maximum yield of 72%, which is comparable to the in vitro folding efficiency for single-chain proinsulin. When the resistance of BPIns to IDE was evaluated in the presence of SeIns, the degradation rate of BPIns became significantly slower than that of BPIns alone. Furthermore, the investigation on the inter-molecular association properties of SeIns and BPIns using analytical ultracentrifugation suggested that SeIns readily forms oligomers not only with its own but also with BPIns. The hypoglycemic effect of SeIns on diabetic rats was observed at a dose of 150 μg/300 g rat. The strategy of replacing the solvent-exposed S-S with Se-Se provides new guidance for the design of long-acting insulin formulations.

[1] Department of Chemistry, School of Science, Tokai University, Kitakaname, Hiratsuka-shi, Kanagawa 259-1292, Japan. [2] Institute of Advanced Biosciences, Tokai University, Kitakaname, Hiratsuka-shi, Kanagawa 259-1292, Japan. [3] Frontier Research Institute for Interdisciplinary Sciences, Tohoku University, 6-3, Aramakiaza Aoba, Aoba-ku, Sendai 980-8578, Japan. [4] Research Center for Bioconvergence Analysis, Korea Basic Science Institute, 162, Yeongudanji-ro, Ochang-eup, Cheongwon-gu, Cheongju-si 28119, Korea. [5] Bio-Analytical Science, University of Science and Technology, 217, Gajeong-ro, Yuseong-gu, Daejeon 34113, Korea. [6] Graduate School of Analytical Science and Technology, Chungnam National University, 99, Daehak-ro, Yuseong-gu, Daejeon 34134, Korea. [7] Research Headquarters, Korea Brain Research Institute, 61, Cheomdan-ro, Dong-gu, Daegu 41068, Korea. [8] Department of Bioengineering, School of Engineering, Tokai University, Kitakaname, Hiratsuka-shi, Kanagawa 259-1292, Japan. [9] Drug Design Laboratory, Graduate School of Medical Life Science, Yokohama City University, 1-7-29 Suehiro, Tsurumi, Yokohama 230-0045, Japan. [10] Division of Nephrology, Endocrinology and Metabolism, Department of Internal Medicine, Tokai University, School of Medicine, 143 Shimokasuya, Isehara, Kanagawa 259-1193, Japan. [11] Institute for Protein Research, Osaka University, Yamadaoka, Suita-shi, Osaka 565-0871, Japan. [12] Institute of Multidisciplinary Research for Advanced Materials, Tohoku University, Aoba-ku, Sendai 2-1-1, Japan. [13] Medical Institute of Bioregulation, Kyushu University, Fukuoka 812-8582, Japan. [14]These authors contributed equally: Kenta Arai, Masaki Okumura, Young-Ho Lee. ✉email: k-arai4470@tokai-u.jp; miwaoka@tokai.ac.jp

nsulin, a small globular peptide hormone (*ca.* 5.8 kDa) with hypoglycemic activity, is composed of two polypeptide chains (A- and B-chains), and its native structure is maintained by three disulfide-bonds (S-S's) formed between the cysteine residues Cys[A6]–Cys[A11], Cys[A7]–Cys[B7], and Cys[A20]–Cys[B19] (Fig. 1a). Slightly more than 100 years have passed since insulin was discovered in a dog's pancreatic homogenate and then such isolated insulins, particularly bovine insulin were used clinically as a hypoglycemic drug for diabetic patients. With the worldwide increase in the number of diabetic patients, much effort is still being devoted to improving the pharmacological effects of insulin formulations. However, the unique double-chain structure of insulin poses a major limitation to its synthetic preparation[1].

On the other hand, the substitution of Cys residues with selenocysteine (Sec; U) in proteins has become a practical method to enhance the in vitro protein folding efficiency and to modify the thermodynamic and kinetic stability of protein structures[2–4]. Although Se and S are homologous elements, non-negligible gaps are often found between wild-type proteins and the selenium analogs[5]. Notably, the reduction potential of diselenide-bond (Se-Se) is significantly lower than that of S-S, rendering the selenium analogs thermodynamically more stable[6]. This often modulates oxidative folding pathways of Sec-substituted proteins and improves folding velocity and productivity of the folded state[7, 8]. Therefore, the similar S-to-Se substitution strategy was also a useful approach to a novel insulin formulation design in order to control the foldability, biological activity, conformation, and metabolic stability. Indeed, two types of artificial insulin analogs (i.e., seleno-insulins), in which one of the three S-Ss is replaced by Se-Se, have already been reported (Fig. 1b, c).

The [C7U[A],C7U[B]] analog (SeIns, Fig. 1b), which was synthesized by our group based on the primary sequence of bovine pancreatic insulin (BPIns), was the first seleno-insulin[9]. While having almost the same folded structure and bioactivity (i.e., phosphorylating activity for Akt and GSK3β in living cells[10]) as those of BPIns, this SeIns analog showed significantly higher resistance in vitro to insulin-degrading enzyme (IDE[11]) ($\tau_{1/2} \approx 8$ h vs. ≈1 h for BPIns), which is widely found in mammalian liver and kidney and responsible for insulin clearance[12]. IDE selectively incorporates a monomeric insulin unit into its catalytic chamber, inducing its partial denaturation and promoting the hydrolysis of internal peptide bonds[13–15]. X-ray crystallography of this SeIns analog revealed that the substituted Se-Se induced an enhancement of the interaction network between Phe[B1] and its neighboring residues (i.e., the N-terminus region in the SeIns B-chain)[9], suggesting a structural stabilization in the monomeric

state, which would at least in part account for the IDE resistance of this SeIns. The latest study on the oligomeric states of human insulin (HIns) indicated that HIns forms oligomers at concentrations higher than 10 μM[16]. This finding may suggest the possibility that SeIns also forms the oligomers and thereby exhibits resistance to IDE degradation.

The second example was a [C6U[A],C11U[A]] analog of HIns reported by Metanis' group (Fig. 1c)[17]. They demonstrated that the substitution of Se-Se for the S-S at Cys6[A]–Cys11[A], which is buried in the hydrophobic core of the molecule, enhanced the thermodynamic stability and decelerated the rates of the hydrolysis by endoproteinase Glu-C and the reductive unfolding with glutathione (GSH). It is of note that the kinetic stability of HIns against degradation was improved by the replacement of the internal S-S with Se-Se.

In the meantime, our report on the IDE resistance of the SeIns analog (Fig. 1b) and its potential applicability as a long-acting insulin formulation have been called into question by Weiss et al.[18]. According to their inspection, our SeIns analog (Fig. 1b), which was synthesized through a different route from ours, possessed no significant resistance to IDE in comparison to BPIns. They also claimed that this analog showed no sustained hypoglycemic effect in an in vivo assay using diabetic model rats. Thus, it was highly demanded to elucidate whether our SeIns analog exhibits really IDE resistance and a sustained hypoglycemic effect in vivo.

In this study, to clarify these aspects, we performed extensive experiments on the stability of our SeIns against IDE degradation. Oligomerizing behaviors and thermal stabilities of the insulin analog were analyzed by analytical ultracentrifugation (AUC)[19] and circular dichroism (CD), respectively, to provide insight into the molecular mechanisms for its remarkable IDE resistance. The dose-dependent hypoglycemic effect was also investigated using diabetic model rats. Based on the results, we hereby argue that the substitution of Se-Se for the solvent-exposed S-S (Cys7[A]–Cys7[B]) is a promising approach to control the stability and association of insulin molecules and thus enhance their sustained efficacy as a hypoglycemic formulation.

## Results and discussion

**Improved oxidative folding of the A- and B-chains into the SeIns analog in up to 72% yield.** Chemical synthesis of insulin by oxidative folding of the A- and B-chains in acceptable yields has posed great problems for decades. More recently, various methods for efficient total synthesis have already been devised[20]. One of the most successful examples is the coupling of the A- and

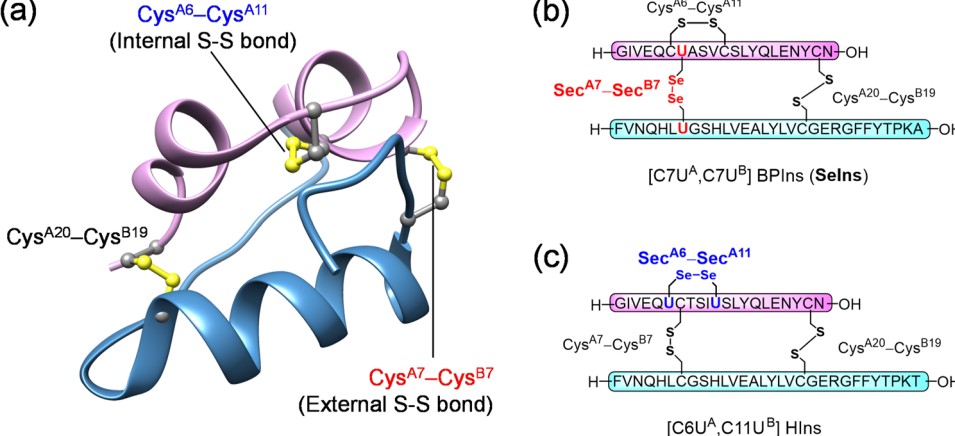

**Fig. 1 Molecular structure of insulins.** Three-dimensional structure of wild-type bovine pancreatic insulin (BPIns; PDB code: 2bn3) (**a**). Primary amino acid sequences of two seleno-insulins having external (**b**) or internal (**c**) Se-Se linkage reported by Arai et al. and Metanis et al. respectively.

B-chains, which have orthogonally protected Cys residues, via the stepwise deprotection of the Cys residues and subsequent S-S formation[21–25]. Using this method, Liu et al. synthesized HIns in a yield of 24% (based on the resin substitution), which is the highest record in insulin synthesis to date[23]. Another example is the application of a single-chain insulin (i.e., a proinsulin mimic), which consists of the A- and B-chains connected with an easily removable linker. After the oxidative folding to gain the native S-S's, the linker was excised chemically or enzymatically to obtain a structurally mature insulin[26–31]. As another simple method, we previously developed native chain assembly (NCA)[32], in which the unprotected native insulin A- and B-chains were directly coupled under optimized conditions (i.e., at pH 10.0 and low temperature in the presence of additives, such as urea and glutathione [GSH/GSSG]). Consequently, the yield of insulin production was increased up to 49% based on the component peptide chains.

Herein, we applied similar NCA conditions for the one-step preparation of the SeIns analog from the suitably protected synthetic A- and B-chains (SeInsA and SeInsB, respectively) (Table 1 and Supplementary Fig. 1). While SeIns was obtained in 27% yield previously (Table 1, entry 1)[9], the yield was dramatically improved to 52% when the conditions of NCA optimized for the preparation of wild-type insulin[32] were applied (Table 1, entry 2). Notably, the folding of SeIns proceeded efficiently even without protein disulfide isomerase (PDI[33–36]), which was required for the effective folding of BPIns (Table 1, entry 3). Furthermore, when SeInsB[SPys,SePys], in which free selenol and thiol groups in the reduced state of SeInsB were protected with 2-pyridylsulfanyl (Pys) groups, was applied as a starting material instead of SeInsB[SeS], the yield was further improved up to about 70%, which is comparable to the in vitro folding yield of proinsulin (Table 1, entry 4)[37]. Again, a similar yield was attained in the absence of PDI, although it took about 6 days to complete the reaction (Table 1, entry 5). These results revealed that NCA is an exquisite folding method for seleno-insulin preparation.

**The coexistence of SeIns enhances the IDE tolerance of BPIns.** In our previous study, the SeIns analog showed about 8-fold higher resistance than BPIns to the peptidyl hydrolysis by human IDE (IDE)[9]. However, Weiss et al. later reported that there was a marginal difference in the IDE resistance between SeIns and BPIns[18]. In these two experiments done independently, the concentrations of the substrate (S; SeIns or BPIns) and the enzyme (E; IDE) were different ([S]/[E] = 2.1 μM/21 nM in our study[9] and 20 μM/100 nM in Weiss's one[18]). Therefore, we investigated how the rate of the IDE-mediated degradation of insulin varies with the substrate and enzyme concentrations. For this, we utilized commercial BPIns (Merck Japan, Japan) as a reference substrate. It is to be noted that the commercial sample was thoroughly purified by RP-HPLC to remove a tiny amount of impurities before use.

Initially, the degradation of SeIns or BPIns was conducted with [S] = 5.0 μM and [E] = 50 nM at 30 °C and pH 8.0 (Fig. 2a). After specific time points, a portion of the reaction solution was acidified to quench the reaction, and immediately analyzed by RP-HPLC (Supplementary Fig. 3a). Amounts of the insulin substrate remaining in the sample were plotted against the reaction time (Fig. 2b). The results clearly showed that SeIns (blue; $\tau_{1/2} \approx 7.6$ h) is significantly more resistant than BPIns (red; $\tau_{1/2} \approx 2.4$ h) to degradation by IDE, being consistent with our previous observation. The apparent first-order rate constants for the enzymatic degradation ($k_{app}$, Fig. 2a) estimated from the single exponential fitting to the experimental data are

summarized in Fig. 2c. The degradation rate of SeIns ($k_{app} = 0.042$ h$^{-1}$; $\tau_{1/2} \approx 17.2$ h) was slower than that of BPIns ($k_{app} = 0.140$ h$^{-1}$; $\tau_{1/2} \approx 5.2$ h) also under the condition employed by Weiss et al. (S/E = 20 μM/100 nM), reinforcing the remarkable IDE resistance of our SeIns (Fig. 2c and Supplementary Fig. 3b, c). It is notable that the rate constants for the degradation of both BPIns and SeIns decreased with an increase in the substrate concentration (Fig. 2c). This feature may be attributed to the intrinsic propensity of insulins to form soluble oligomers in aqueous solutions even in the absence of Zn$^{2+}$ that is an intermolecular association enhancer for insulins[38], as observed for HIns[16]. In line with this, the catalytic chamber of IDE has the spatial capacity to accept a monomeric insulin as a substrate (Fig. 2a)[14], and hence the oligomer formation of insulin substrates could significantly affect the degradation rate by IDE. Another interesting and important observation is that the degradation was intrinsically much faster for BPIns than for SeIns (Fig. 2b, c).

These features can reasonably be explained by the preequilibrium model (Fig. 3). In this model, two factors of the monomer units are involved, i.e., the easiness for oligomer formation and the resistance to IDE degradation. The net decay rate for SeIns would become slower when the monomers form oligomers at higher ratios ($K_d^{SeIns} < K_d^{BPIns}$; $K_d$ represents an equilibrium constant for the dissociation of the oligomer) and/or the monomer is more resistant against IDE degradation ($k_{deg}^{SeIns} < k_{deg}^{BPIns}$) (Fig. 3, models (a) and (b)). To assess the relative importance of these two factors, we performed the degradation experiments for the mixture of SeIns and BPIns (Fig. 2d). When the 1:1 mixture was employed at the total concentration of 5.0 μM, BPIns and SeIns were degraded at almost the same rates or slightly faster for BPIns (i.e., $k_{app}^{BPIns} = 0.097$ h$^{-1}$ vs $k_{app}^{SeIns} = 0.095$ h$^{-1}$ [Fig. 2c]). Importantly, the total decay rate of the insulin mixture was comparable to that observed for SeIns alone (Fig. 2b, green), indicating that the coexistence of SeIns enhanced the apparent IDE resistance of BPIns. This is presumably because hetero-oligomers are readily formed between BPIns and SeIns, from which BPIns and SeIns monomers will reversibly dissociate with similar probability under an equilibrium constant $K_d^{mix}$ and be degraded by IDE (Fig. 3, model (c)). Similar results were observed when the mixing ratio of BPIns and SeIns was changed to 1:4. (Fig. 2b, c and Supplementary Fig. 4). Interestingly, even when the opposite mixing ratio, i.e., 4:1 (BPIns:SeIns), was applied, the degradation of BPIns was significantly decelerated by the coexistence of SeIns (Fig. 2b, c and Supplementary Fig. 4). The results of the competition experiments strongly suggested that the observed IDE resistance of SeIns is mainly due to the easier formation of the oligomers rather than the enhanced IDE resistance of monomer units.

**SeIns readily forms oligomers not only with its own but also with BPIns.** To verify the preequilibrium model (Fig. 3), the oligomerizing behavior of SeIns was investigated by analytical ultracentrifugation (AUC) at the various concentrations ([SeIns] = 5, 10, and 20 μM) (Fig. 4). It should be noted that the lowest concentration of insulins that was applicable in the equipment was 5 μM. At 5 μM, a single histogram peak was observed for BPIns in the area of sedimentation coefficient under ca. 1 S or less, which corresponds to monomeric insulin (Fig. 4a)[16]. On the other hand, an additional peak with a sedimentation coefficient around 2 S, which corresponds to the dimer[16], was observed for SeIns at the same concentration, indicating that SeIns can form dimers at concentrations less than 5 μM (Fig. 4b). Importantly, a histogram peak of dimers was more clearly observed for the mixed solution ([BPIns] = [SeIns] = 2.5 μM) (Fig. 4c), suggesting that

**Table 1 Optimization of two-chain oxidative folding conditions for preparation of SeIns.**

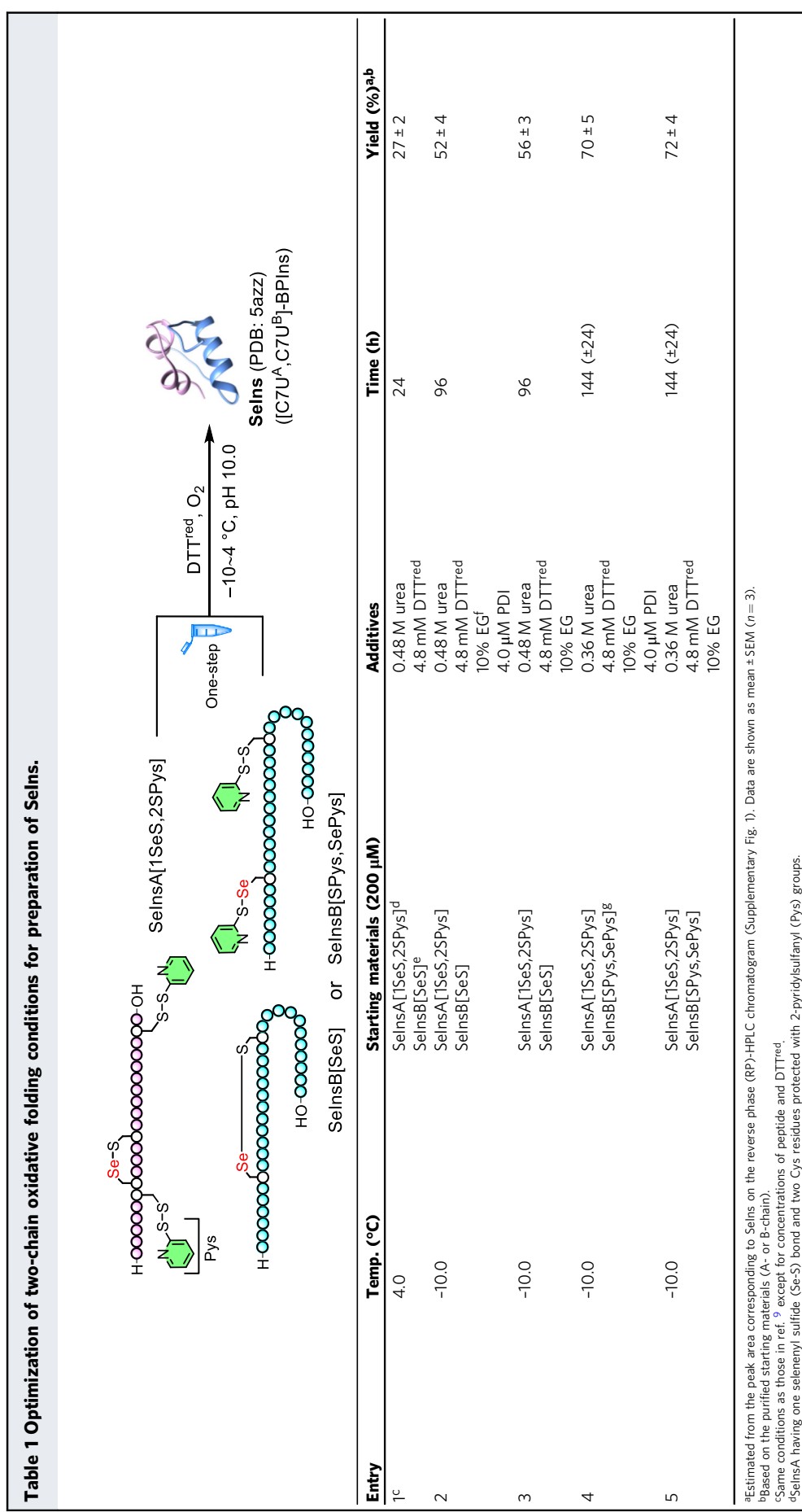

| Entry | Temp. (°C) | Starting materials (200 μM) | Additives | Time (h) | Yield (%)[a,b] |
|---|---|---|---|---|---|
| 1[c] | 4.0 | SeInsA[1SeS,2SPys][d] SeInsB[SeS][e] | 0.48 M urea 4.8 mM DTT[red] | 24 | 27 ± 2 |
| 2 | −10.0 | SeInsA[1SeS,2SPys] SeInsB[SeS] | 0.48 M urea 4.8 mM DTT[red] 10% EG[f] 4.0 μM PDI | 96 | 52 ± 4 |
| 3 | −10.0 | SeInsA[1SeS,2SPys] SeInsB[SeS] | 0.48 M urea 4.8 mM DTT[red] 10% EG | 96 | 56 ± 3 |
| 4 | −10.0 | SeInsA[1SeS,2SPys] SeInsB[SPys,SePys][g] | 0.36 M urea 4.8 mM DTT[red] 10% EG 4.0 μM PDI | 144 (±24) | 70 ± 5 |
| 5 | −10.0 | SeInsA[1SeS,2SPys] SeInsB[SPys,SePys] | 0.36 M urea 4.8 mM DTT[red] 10% EG | 144 (±24) | 72 ± 4 |

[a]Estimated from the peak area corresponding to SeIns on the reverse phase (RP)-HPLC chromatogram (Supplementary Fig. 1). Data are shown as mean ±SEM (n = 3).
[b]Based on the purified starting materials (A- or B-chain).
[c]Same conditions as those in ref. 9 except for concentrations of peptide and DTT[red].
[d]SeInsA having one selenenyl sulfide (Se-S) bond and two Cys residues protected with 2-pyridylsulfanyl (Pys) groups.
[e]SeInsB having one Se-S bond.
[f]Ethylene glycol.
[g]SeInsB having Cys and Sec residues protected with Pys groups (Supplementary Fig. 2).

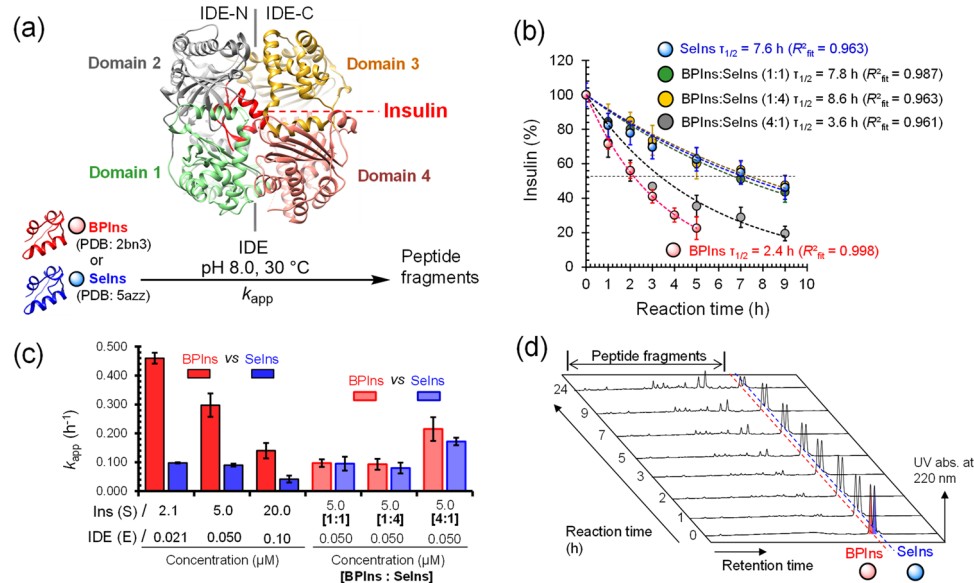

**Fig. 2 Enzymatic digestion of insulins by human insulin-degrading enzyme (IDE). a** Degradation of BPIns or SeIns catalyzed by IDE (PDB code: 2wby)[14]. Monomeric insulin bound into the catalytic chamber of IDE is shown in red. **b** Time course of insulin degradation observed by HPLC analyses (Supplementary Figs. 3a and 4). Reaction conditions were [inslins]$_0$ = 5.0 μM and [IDE] = 50 nM (S/E = 100:1) in 0.1 M Tris-HCl at pH 8.0 and 30 °C. Decay rates in the competitive experiments using SeIns-BPIns mixture are shown as total insulin degradation. Data are shown as mean ± SEM (n = 3). **c** Comparison of apparent first-order rate constant ($k_{app}$) for degradation of insulins. Data are shown as mean ± SEM (n = 3). The values were estimated by a single exponential fitting of time course data for insulin degradation with the equation: %insulin = 100($e^{kt}$). Data for S/E = 2.1 μM/21 nM were estimated from previous results in ref. [9]. **d** HPLC charts obtained from digestion experiment of SeIns-BPIns mixture by IDE at pH 8.0 and 30 °C. Mixture of insulins (5.0 μM [BPIns:SeIns = 1:1]) were incubated with IDE (50 nM) under the same conditions as those in (**b**).

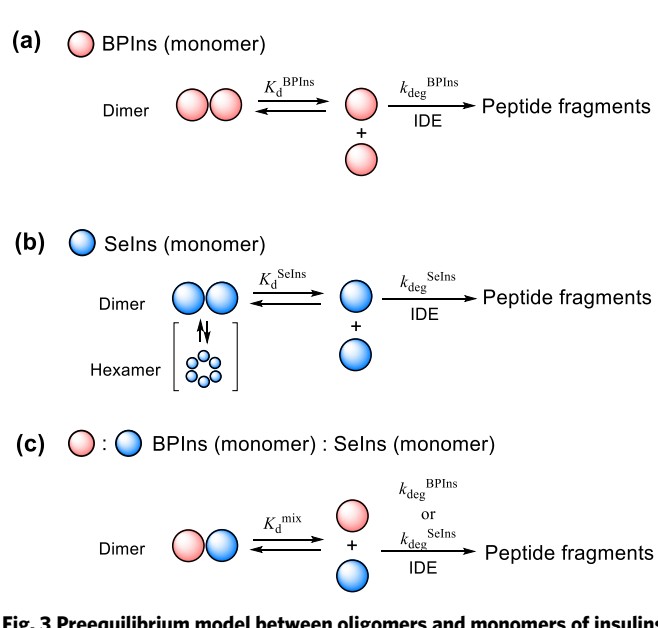

**Fig. 3 Preequilibrium model between oligomers and monomers of insulins during IDE degradation. a** Proposed degradation pathways of BPIns including reversible dissociation of BPIns oligomers with an equilibrium constant ($K_d^{BPIns}$) and irreversible degradation of monomeric BPIns with a pseudo first-order rate constant ($k_{deg}^{BPIns}$). **b** Proposed degradation pathways of SeIns including reversible dissociation of SeIns oligomers with an equilibrium constant ($K_d^{SeIns}$) and irreversible degradation of monomeric SeIns with a pseudo first-order rate constant ($k_{deg}^{SeIns}$). **c** Proposed degradation pathways of BPIns–SeIns mixture including reversible dissociation of BPIns–SeIns oligomers with an equilibrium constant ($K_d^{mix}$) and irreversible degradation of monomeric BPIns or SeIns with a pseudo first-order rate constant ($k_{deg}^{BPIns}$ or $k_{deg}^{SeIns}$).

that the oligomerizing capability of BPIns is enhanced by the coexistence of SeIns. At higher concentrations, the histogram for the dimer was increased for SeIns and the SeIns-BPIns mixture, (Fig. 4b, c, 10 and 20 μM), whereas for BPIns, a broadening of the corresponding population was observed (Fig. 4a, 10 and 20 μM). As for SeIns, a population corresponding to a hexamer (sedimentation coefficient ca. 4 S[16]) was also observed at 20 μM, despite the absence of Zn$^{2+}$. Of note, the sum of histograms of 5 μM BPIns and 5 μM SeIns (Fig. 4d, bottom panel) was obviously different from the histogram distribution of the mixture sample comprising 5 μM BPIns and 5 μM SeIns (Fig. 4c, 10 μM). This was the same case with the comparison between the mixture samples of 10 μM each (i.e., Fig. 4c, 20 μM vs. Fig. 4d, top panel). These observations suggest that a major portion of oligomers observed in the mixture (Fig. 4c) is derived from hetero-oligomer formation between SeIns and BPIns. In addition, the observed concentration-dependent oligomerizing behaviors of BPIns, SeIns, and the mixture are consistent with their behaviors during the degradation by IDE, supporting the preequilibrium model between oligomers and monomers (Fig. 3). The remarkable capability of SeIns to form oligomers may be due in part to the fact that the solvent-exposed Se-Se bond is more polarized than the S-S bond and consequently assists in the association between insulin molecules.

**The oligomerizing properties of SeIns stabilize the insulin molecular structure.** Subsequently, the structural properties of SeIns and BPIns in solution were investigated using CD spectroscopy. As presented in Supplementary Fig. 6, the far-UV CD spectra of SeIns and BPIns showed two clear minima at 208 and 222 nm, indicating that the two types of insulins are predominantly composed of α-helical structures. Although the CD spectra of SeIns and BPIns were broadly similar to each other, the

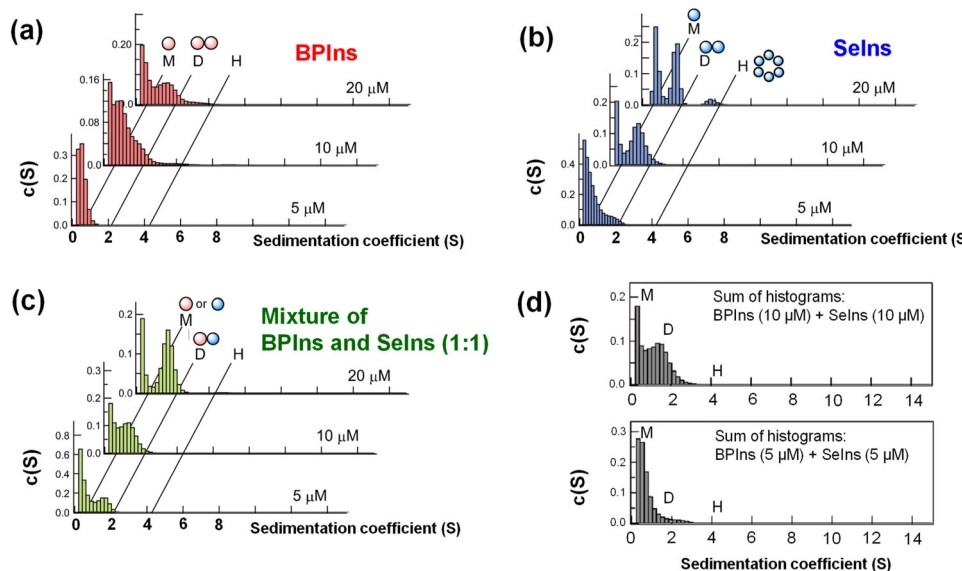

**Fig. 4 Apparent sedimentation coefficient histograms of insulin monomer and oligomers estimated by analytical ultracentrifugation (AUC) (Supplementary Fig. 5).** Symbols M, D, and H in the panels represent monomer, dimer, and hexamer, respectively. BPIns (**a**), SeIns (**b**), or their mixture (**c**, 1:1) were dissolved in 0.1 M Tris-HCl buffer solution at pH 8.0 and a variety of concentrations (5, 10, and 20 μM). The sums of the histograms obtained from the single-component analyses, i.e., BPIns (5 μM in **a**) + SeIns (5 μM in **b**) [bottom] and BPIns (10 μM in **a**) + SeIns (10 μM in **b**) [top], are shown in (**d**).

| Table 2 Thermodynamic parameters of heat denaturation of insulins obtained using CD spectroscopy[a]. | | | |
|---|---|---|---|
| **Sample[b]** | $T_m$ **(°C)[c]** | **ΔH (kcal mol$^{-1}$)[d]** | **ΔC$_p$ (kcal K$^{-1}$ mol$^{-1}$)[e]** |
| SeIns | 72.6 ± 0.2 | 24.5 ± 6.1 | 0.5 ± 0.2 |
| BPIns | 58.4 ± 1.3 | 19.6 ± 1.2 | 0.4 ± 0.1 |

[a]Three independent experiments were performed and values are presented as mean ± SEM ($n = 3$).
[b]Insulins (20 μM) were dissolved in 20 mM sodium phosphate buffer (pH 7.5) containing 1 M Gdn-HCl.
[c]Melting temperature ($T_m$).
[d]Enthalpy change (ΔH).
[e]Heat capacity change (ΔC$_p$) of thermal unfolding calculated using fit analyses (see "Methods" for the details).

CD intensity and profile were distinct, suggesting a possibility of a structural difference. The prediction of the secondary structure content using the BeStSel algorithm[39] revealed that the α-helix content of SeIns (32.7%) was slightly, but, appreciably lower than that of BPIns (36.5%), which is consistent with the CD spectra obtained by Weiss and coworkers[18]. Notably, our previous structural study in crystals demonstrated that SeIns monomer has the same three-dimensional structure as that of BPIns monomer, and thus the content of the secondary structure element does not vary depending on the type of insulin monomers[9]. Therefore, the decrease in the α-helix content of SeIns compared to BPIns may be attributable to their different oligomerization states in solution.

Next, the conformational stability of SeIns and BPIns was examined using the thermodynamic analysis of the temperature-dependent far-UV CD spectra from 5 to 110 °C. Here, to slightly destabilize the molecular structures of insulins, Gdn-HCl, a well-known chemical denaturant, was added to the sample solutions because the thermal unfolding of SeIns and BPIns without Gdn-HCl was not completed in the temperature range examined (Supplementary Fig. 7). In the presence of 1 M Gdn-HCl, the CD intensities of SeIns and BPIns at 222 nm increased from ~50 and ~40 °C, respectively, and both of them reached a plateau at

~90 °C (Supplementary Fig. 8a, b), indicating that the complete disruption of secondary structures of SeIns and BPIns at high temperatures. Although the refolding experiments by cooling down the samples from 110 °C suggested that the thermal unfolding of SeIns and BPIns was irreversible (Supplementary Fig. 8c, d), with the assumption of the reversible unfolding, fitting analyses of the thermodynamic stability of SeIns and BPIns were performed (Table 2). The apparent melting temperature ($T_m$) of SeIns was 72.6 ± 0.2 °C, higher than that of BPIns (58.4 ± 1.3 °C). The apparent values of the enthalpy change (ΔH) for SeIns and BPIns unfolding were 24.5 ± 6.1 and 19.6 ± 1.2 kcal mol$^{-1}$, respectively. Collectively, these results suggested that SeIns, which can form higher oligomers than BPIns at the concentrations used (Fig. 4a, b; 20 μM), has a higher structural stability than BPIns.

**SeIns exerts a sustained hypoglycemic effect**. It is interesting to see how the unique oligomerizing property and protease resistance of the SeIns analog affect its pharmacological action as an insulin formulation. Therefore, we next investigated the hypoglycemic effect of SeIns by in vivo experiments using normal rats and pathological rat models, for which diabetes was induced by streptozotocin (STZ) (Fig. 5). First, insulin lispro (Humalog®[40–42]) as a rapid-acting insulin formulation, synthetic HIns[32], synthetic BPIns[32], or SeIns was dissolved in normal saline solution (10 units/mL), and administered to rats through a subcutaneous (s.c.) injection at 15 μg/300 g rat. The changes in blood glucose levels were measured by using the i-STAT 1® monitoring system. In normal rats, the glucose levels became minimal at approximately 1 h post-dose for all samples, and there was no significant difference in the pharmacological action among the insulin analogs, although insulin lispro showed slightly faster recovery of the blood glucose level (Fig. 5a). In diabetic rats, a decrease in the glucose level and its recovery were observed in a similar manner as that observed in normal rats (Fig. 5b). Thus, no significant long-acting behavior was observed for SeIns. Similar results were reported by Weiss et al. after administration of SeIns (10–15 μg/300 g rat)[18]. On the other hand, when the same experiment was conducted using a 10-fold higher dose of HIns and SeIns (150 μg /300 g rat) and the blood glucose levels were

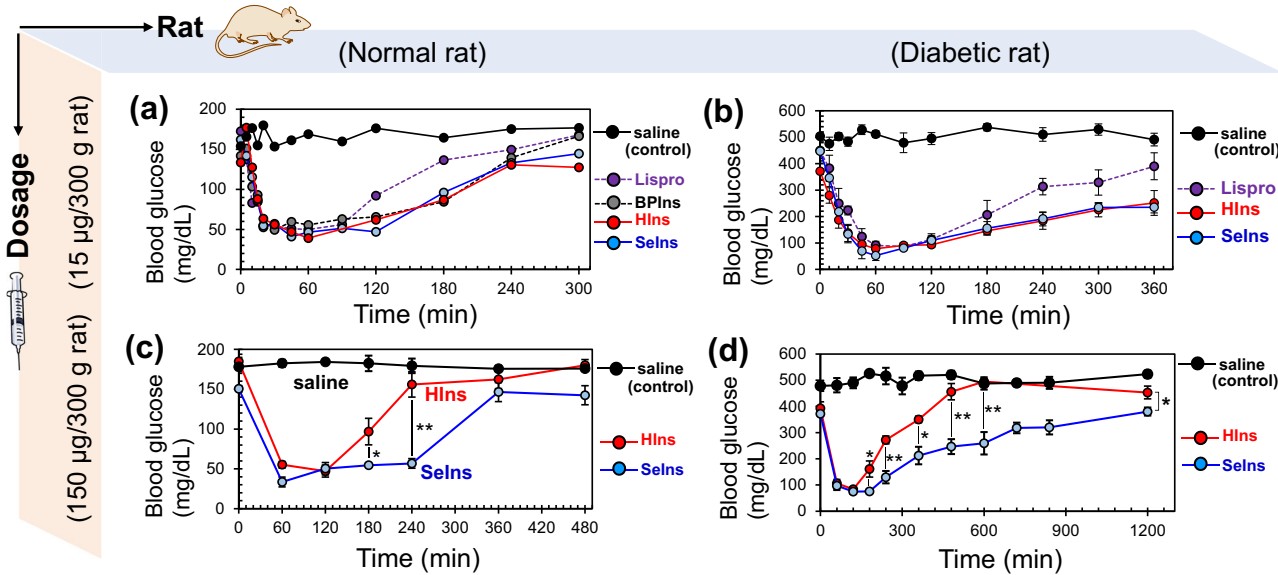

**Fig. 5 In vivo study on the hypoglycemic effect of insulins using normal and diabetic model rats. a** Time course of blood glucose level in normal rat following subcutaneous (s.c.) injection of insulin samples (10 units/mL of saline). The dosage of insulins was 15 μg/300 g rat. The experiments with saline and insulins were repeated four and two times, respectively, and good repeatability was confirmed. Data are shown as the average. **b** Time course of blood glucose level in STZ-induced diabetic model rat following s.c. injection of insulin samples (10 units/mL of saline). The dosage of insulins was 15 μg/300 g rat. Data are shown as mean ± SEM ($n = 3$ [insulins] or 5 [saline]). **c** Time course of blood glucose level in normal rat following s.c. injection of insulin samples (100 units/mL of saline). The dosage of insulins was 150 μg/300 g rat. Data are shown as mean ± SEM ($n = 5$ [insulins] or 4 [saline]). **d** Time course of blood glucose level in STZ-induced diabetic model rat following s.c. injection of insulin samples (100 units/mL of saline). The dosage of insulins was 150 μg/300 g rat. Data are shown as mean ± SEM ($n = 4$ [HIns] or 5 [SeIns and saline]). The blood glucose level was monitored by an i-STAT 1 analyzer with cartridge 6+. The symbols * and ** represent $p < 0.05$ and $p < 0.01$, respectively. The $p$-value was obtained from a $t$-test.

measured by i-STAT 1® analyzer using a minimum volume of the blood collected from the tail vein, the blood glucose reached the lowest level in 1–3 h after the administration and a long-acting behavior of SeIns was clearly observed in both normal (Fig. 5c) and diabetic model rats (Fig. 5d). It should be noted that difference in the survival rates was not observed between the rats treated with SeIns and HIns (Supplementary Table 1), indicating that the short-term toxicity of SeIns is low in the applied range of SeIns dose (15–150 μg/ 300 g rat). Needless to say, it is also important to investigate the long-term effects of SeIns in future studies because daily administration of insulin is required for diabetic patients.

The long-acting behavior of SeIns observed in vivo can be rationalized by considering that the oligomers of SeIns formed at a high concentration would be gradually secreted from the subcutaneous region to the bloodstream through the slow dissociation to the monomers compared to BPIns. Alternatively, SeIns would form the oligomers and/or hetero-oligomers with wild-type insulin, thereby slowing down its degradation by proteases in the blood and/or by IDE during the hepatic clearance. To seek other possibilities, the stability of SeIns against thiol-based reducing factors existing in the blood was compared with that of BPIns (see Supplementary Methods for experimental details). Reductive degradation (unfolding) of insulin is initiated by reductive cleavage of the solvent-exposed S-S (Cys7[A]–Cys7[B])[32]. The replacement with thermodynamically more stable Se-Se may decelerate the reductive degradation. However, the rate of the reduction of SeIns by glutathione (GSH) was almost the same as that of BPIns (Supplementary Fig. 9), suggesting that the long-acting property of SeIns is not attributed to its resistance to thiol-based reducing factors. Consistently, previous works demonstrated that the thermodynamically more stable Se-Se is kinetically easier to cleavage than S-S[43, 44]. On the contrary, [C6U[A],C11U[A]]HIns was found to be more resistant to the reduction by GSH than the wild-type insulin[17]. Thus, the

reductive degradation of seleno-insulins is greatly affected by the position of the Se-Se replacement.

In addition, to elucidate the relative stability of SeIns in the blood, the degradation rates of SeIns and BPIns were directly compared in human serum according to the previous method[45, 46] with slight modifications (see Supplementary Methods for experimental details). The sample (BPIns or SeIns; 10 μM) was exposed to human serum at 37 °C, and the remained insulin in the sample solution after a certain period of time was estimated by RP-HPLC. As a result, both SeIns and BPIns were gradually degraded in a manner of pseudo first-order reaction (Supplementary Fig. 10), and the degrading rate of SeIns in human serum ($k_{obs} = 0.015$ h$^{-1}$, $\tau_{1/2} = 46$ h) was slightly but distinctly slower than that of BPIns ($k_{obs} = 0.020$ h$^{-1}$, $\tau_{1/2} = 35$ h). The result indicates that the substitution of Se-Se for the Cys7[A]–Cys7[B] S-S enhances the stability in the blood. This would be advantageous for SeIns as instability in the blood is one of the obstacles in the development of peptide-based injectable formulations. Nonetheless, the observation also indicates that the stability of SeIns in the blood would not be a major factor for the long-acting property because the difference in the degradation rate between BPIns and SeIns was not enough to explain the remarkable difference in the blood glucose level between mice treated with BPIns and SeIns. Thus, it is surmised that the long-acting property of SeIns is exerted by its resistance to degradation during the hepatic insulin clearance due to the improved IDE tolerance and/or the delayed diffusion of the insulin monomers into the bloodstream after the s.c. administration. To propose more detailed and accurate molecular mechanisms for the long-acting property of SeIns, further pharmacokinetic analyses are necessary.

## Conclusion
In this study, applying the NCA conditions[32], which were previously optimized for the preparation of wild-type insulins, to

SeIns synthesis, the yield of chain-coupling reached up to 72% yield which is comparable to that of oxidative folding of proinsulin. Considering the application to industrial production, the chemical synthesis of insulin would be more costly than the current protein expression method using recombinant DNA. However, using the chemical method, such as the double-chain assembly optimized in this study, the introduction of site-specific mutations with natural and non-natural amino acid residues should be easier, hence it would help speed up the structural screening for the development of insulin formulation candidates. On the other hand, the superior resistance of SeIns to IDE was reconfirmed at various substrate concentrations. Furthermore, it was revealed that the IDE resistance of SeIns is largely explained by its own remarkable associating property to form soluble homo-oligomers and hetero-oligomers with the wild-type insulin, presumably due to the presence of a more polarizable Se-Se than S-S on the insulin surface. The contribution of the enhanced conformational stability of SeIns as revealed by X-ray crystallography[9] and thermal denaturation experiments (Table 2) cannot be ignored because the degradation of SeIns was slightly slower than that of BPIns in the competitive IDE-degradation experiments. Moreover, it was also observed that SeIns exhibits a long-acting property at a dose of 150 µg/300 g rat in the study using diabetic model rats.

In the history of the development of insulin formulations for diabetic patients, long-acting insulin, which supplements basal insulin secretion, has been developed on the basis of inhibiting dissociation of bioactive monomeric insulin, exploiting insoluble precipitate formation after subcutaneous injection (e.g., glargine[47–49]), complex formation with albumin in the blood via higher fatty acids conjugated with the insulin molecule (e.g., detemir[50–52]), and higher-order complex formation with zinc ions (e.g., degludec[53–56]). In addition, insulin icodec, a once-weekly formulation, that is conjugated with C20 fatty diacid (icosanedioic acid) can exhibit an ultra-long-acting property due to its extremely strong association with albumin and is undergoing clinical trials for use for new basal insulin therapy[57–59]. Although the sustained potency of our SeIns would be mild compared to this insulin analog, the present study may provide another approach to the design of long-acting insulin formulations, in which the resistance to proteases, including IDE, is enhanced by the S to Se substitution. Considering the possible formation of hetero-oligomers, SeIns may be administered as the mixture with other insulin formulations to control the sustained effect. Thus, the development of SeIns analogs is a promising strategy for the design of new long-acting insulins, although additional basic data, such as long-term toxicity tests, need to be accumulated in preclinical studies. Further exploration and structural optimization of SeIns-based formulations will be necessary.

## Methods

**Double-chain combination of seleno-insulin via native chain assembly**. SeInsA[1SeS,2SPys] and SeInsB[SeS] were prepared as per our previous report[9]. Synthetic protocols for SeInsB[SPys,SePys] were described in Supplementary Methods. A solution of SeInsA[1SeS,2SPys] (200 nmol) in a sodium bicarbonate buffer (25 mM, pH 10.0, 180 µL) containing urea (1.0 M) and EDTA (1 mM) was mixed with a solution of SeInsB ([SeS] or [SPys,SePys]) (200 nmol) in the same buffer solution (180 µL). The resulting mixture (360 µL) was mixed with 100 µM (or none) PDI (40 µL) and 34.3 mM DTT$^{red}$ (140 µL), which were prepared by a sodium bicarbonate buffer solution (25 mM, pH 10.0) without urea. The obtained sample solution was then diluted by the addition of the same buffer solution without urea (360 µL)

and ethylene glycol (EG, 100 µL). The prepared sample solution (1000 µL) was incubated at −10 or 4 °C. To monitor the reaction progress, small aliquots (5 µL) were taken from the mixture, quenched by the addition of aqueous 2-aminoethyl methanethiosulfonate (AEMTS) solution (7 mg mL$^{-1}$. 200 µL), diluted with 0.1% TFA (835 µL) in water, and analyzed by the HPLC system equipped with a sample solution loop (1 mL) and a Tosoh TSKgel ODS-100V φ 4.6 ×150 mm RP column (Tosoh Corporation, Japan), which was equilibrated with a 80:20 (v/v) mixture of TFA (0.1%) in water (eluent A) and TFA (0.1%) in CH$_3$CN (eluent B) at a flow rate of 1 mL min$^{-1}$. After injection of the sample solution (1.0 mL) onto the HPLC system, a solvent gradient was applied: a ratio of eluent B linearly increased from 20 to 36% in 0–15 min, from 36 to 39% in 15–20 min, from 39 to 40% in 20–23 min. The products during the folding were detected by the absorbance at 280 nm. After completion of the reaction, generated SeIns was isolated by the same HPLC system and lyophilized to obtain white powder. The products were characterized by HPLC retention time, MALDI-TOF-MS and amino acid analysis (AAA). The percentage of yields is summarized in Table 1.

**Degradation of insulins by human insulin-degrading enzyme**. Human insulin-degrading enzyme (IDE) was purchased from Bon Opus Biosciences. Insulin analog (5.0 µM) was mixed with IDE (50 nM, 1:100 E/S) and incubated at 30 °C in buffer containing 0.1 M Tris-HCl (pH 8.0). Similarly, a high concentration sample solution, in which 20 µM of insulin in 0.1 M Tris-HCl (pH 8.0) was treated with IDE (100 nM, 1:200 E/S), was prepared. The resulting sample solution was incubated at 30 °C without mixing. After specific time points, aliquot was removed and quenched by the addition of 1 M HCl. In the degradation assay for the single component (i.e., BPIns or SeIns), the samples were analyzed by the HPLC system equipped with a COSMOSIL 5C18-AR-II φ 4.6 ×150 mm RP column (Nacalai Tesque, Japan), which was equilibrated with a 95:5 (v/v) mixture of TFA (0.05%) in water (eluent A') and TFA (0.05%) in CH$_3$CN (eluent B') at a flow rate of 1 mL min$^{-1}$, and eluted using a linear gradient of eluent B' from 5% to 65% at a rate of 1% min$^{-1}$, with detection at an absorbance of 220 nm. Similarly, in the degradation assay for BPIns–SeIns mixture (5 µM; 1:1, 1:4, or 4:1), the samples were analyzed by HPLC equipped with a Tosoh TSKgel ODS-100V φ 4.6 × 150 mm RP column (Tosoh Corporation, Japan) with detection at an absorbance of 220 nm. After injection of the sample solution onto the HPLC system, a solvent gradient was applied: a ratio of eluent B' linearly increased from 10 to 30% in 0–20 min and from 20 to 54% in 20–100 min.

**Analytical ultracentrifugation analysis**. All samples were prepared with 0.1 M Tris-HCl buffer solution (pH 8.0) as a solvent. The analytical ultracentrifugation (AUC) experiments were performed using an Optima XL-I analytical-ultracentrifuge (Beckman Coulter, USA). The samples were precentrifuged at 3000 rpm for all sample solutions. The collected supernatants were then analyzed by sedimentation velocity-AUC at 60,000 rpm and at 20 °C. The experiments were carried out using the absorbance at 230 nm. The distribution of the sedimentation coefficient was analyzed using SEDFIT[60]. The solvent parameters used in the analysis were calculated using SEDNTERP. The experiments were repeated three times, and the good repeatability was confirmed.

**Circular dichroism analysis**. Far-UV circular dichroism (CD) spectra were measured with a JASCO J-1500 spectrophotometer (JASCO, Japan) using a quartz cuvette with a 0.1 cm path length.

Samples were prepared using 20 µM of insulins in 20 mM sodium phosphate buffer (pH 7.5). The concentration of insulins was determined by measuring the UV absorbance at 280 nm ($\varepsilon_{280} = 5840\ \mathrm{M}^{-1}\ \mathrm{cm}^{-1}$). 8 scans of individual samples were accumulated and averaged. The bandwidth and scan rate were 1 and 200 nm min$^{-1}$, respectively. CD signals between 195 and 250 nm were expressed as the mean residue ellipticity, $[\theta]$ (degrees cm$^2$ dmol$^{-1}$)[61]. The content of the secondary structures was estimated using the β-structure selection (BeStSel) algorithm[39]. For the thermal scans of insulins, samples were prepared using 20 µM of insulins in 20 mM sodium phosphate buffer (pH 7.5) with and without 1 M Gdn-HCl. Far-UV CD spectra were acquired every 5 °C from 5 to 100 or 110 °C at a heating rate of 1 °C min$^{-1}$. Then, the CD signal at 222 nm was monitored for cooling of insulins from 110 to 5 °C with a rate of 10 °C min$^{-1}$. A RW-0525G Low Temperature Bath Circulator (Lab Companion, Republic of Korea) was used for temperature regulation.

The melting temperature ($T_m$), the enthalpy change ($\Delta H$), and the heat capacity change ($\Delta C_p$) for the thermal denaturation of insulins were obtained by a regression analysis under the assumption of a two-state transition between folded and unfolded insulins using the following equation (Eq. (1))[62]:

$$\theta = \frac{(a-c)+(b-d)T}{1+\exp(-\frac{\Delta H(T_m)}{R}\left(\frac{1}{T}-\frac{1}{T_m}\right)+\frac{\Delta C_p}{R}(\frac{T_m}{T}-1+\ln\frac{T}{T_m}))}+(c+dT)$$

(1)

where $\theta$ is the CD signal intensity. $a + bT$ and $c + dT$ are the pre- and post-unfolding baselines, respectively. $T$ and $R$ represent temperature and gas constant, respectively. $\Delta H(T_m)$ is the enthalpy change at $T_m$.

**Rat experiments**. The animal experiment protocol was approved by the Animal Facility Use Committee (Permit No.: 194019) and conducted in accordance with the Tokai University Animal Experiment Regulations. Five-week-old male Sprague-Dawley rats (CLEA Japan Inc., Tokyo, Japan) were intravenously injected with streptozotocin (60 mg/kg), and individuals with blood glucose levels of ≥300 mg/dL after 1 week were used as models of diabetes (STZ rats) for the following experiments. Insulins were dissolved in normal saline solution (10 or 100 Units/mL), and administered to normal and STZ rats through a subcutaneous (s.c.) injection at 15 or 150 µg (1.3 or 13 Units)/300 g rat. The changes in blood glucose levels were measured by an i-STAT 1® analyzer with cartridge 6+ (Abbott Lab. Chicago, IL, USA) using a drop of blood from the tail vein.

**Statistical analyses**. Results obtained from rat experiments represent the mean (± standard deviation) of at least three samples. Comparisons between the two groups were performed using Student's *t*-test. *P*-values of <0.05 were considered statistically significant.

**Reporting summary**. Further information on research design is available in the Nature Portfolio Reporting Summary linked to this article.

## Data availability
The authors declare that the data supporting the findings of this study are available within the article and the Supplementary Information as well as from the authors upon reasonable request.

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

## Acknowledgements

We gratefully acknowledge Prof. Luis Moroder (Max Planck Institute of Biochemistry) for reading our paper and providing valuable suggestions to improve the manuscript. We also thank Dr. Yunseok Heo (Korea Basic Science Institute) for the technical support in structural analyses of insulin analogs. This work was supported by the PMAC for Private School of Japan (The Science Research Promotion Fund [to K.A.]), Research and Study Project of Tokai University, Educational System General Research Organization (to K.A., M.O., H.K., K.S., M.T., and M.I.), JSPS KAKENHI (Grant Number: 23K04933 [to K.A.], 22H02205 [to M.O.]), JSPS Grant-in-Aid for Transformative Research Areas (B) (Grant Number: JP21H05095 [to M.O.]), Japan Science and Technology Agency FOREST Program (Grant Number: JPMJFR201F [to M.O.]), the Takeda Science Foundation (to K.I. and M.O.), the Mochida Memorial Foundation for Medical and Pharmaceutical Research (to M.O.), the Naito Foundation (to M.O.), the Uehara Memorial Foundation (to M.O.), the Terumo Life Science Foundation (to M.O.), National Research Foundation of Korea grant funded by the Korean government (Grant numbers: NRF-2019R1A2C1004954 and NRF-2022R1A2C1011793 [to Y.-H.L.]), National Research Council of Science & Technology grant funded by the Korean government (Grant number: CCL22061-100 [to Y.-H.L.]), and KBSI fund (Grant numbers: C220000, C230130, and C390000 [to Y.-H.L.]).

## Author contributions

K.A. and M.I. developed the idea for this project and initiated it. In addition, these authors designed the experimental parts. K.A. and H.K. synthesized SeIns and HIns under the supervision of M.I. and H.H. M.O. and K.A. evaluated the IDE resistance of insulin analogs. M.O. and Y.-H.L. conducted AUC analysis of insulin analogs under the supervision of K.M. and S.-Y.P. K.S. and M.T. performed rat experiments. M.O., Y.L., and Y.-H.L. performed structural analyses of insulin analogs under the supervision of K.I. K.A. and M.I. prepared the manuscript, which was edited by all authors.

## Competing interests

The authors declare no competing interests.
