## [Peer Review File · Communications Chemistry]

Reviewers' comments:

Reviewer #1 (Remarks to the Author):

The present study builds on previous reports by the group on the synthesis and characterization of seleno-insulin and other insulins. The paper reports an improved synthesis of the original seleno-insulin variant, where the main part is the optimized two-chain oxidative folding.

It is an interesting study and performed at a high level.

The abstract indicates that seleno-insulin is a single insulin variant, while the main text mentions the other seleno-insulins. It would be good if the authors could be more clear about that seleno-insulins is a class of compounds (insulins), where the authors reported the first of these.

It also includes an analytical ultracentrifugation study on the oligomerization the synthesized seleno-insulin. The authors hypothesize that they can observe the effect of formation of hetero-oligomers of insulins. However, while this may be correct, they do not have direct evidence for this.

It also includes a careful study of resistance to processing by IDE, in response to a paper by Weiss et al, which contradicted their original results. The results reported here seem trustworthy. Could the authors more explicitly discuss why their findings differ from Weiss et al.?

The authors comment that their seleno-insulin likely has low short-term toxicity. However, there seems to be no comment on the possible toxicity with longer-term use. This is an important question, as T1D is chronic and requires daily injections. The authors really should comment on this.

The authors state that this work could contribute to the development of longer-acting insulins. While the results on the inhibitory effect on IDE (insulin degrading enzyme), which already was part of the original publication, is interesting and very relevant, there is a lot more to making a long-acting insulin.

The seleno-insulin was assembled by chemical synthesis, whereas the approved long-acting insulins are all made by expression. Chemical synthesis would be more costly, as multi-ton amounts are required. The author should consider this.

Also, the long-acting effect achieved with their seleno-insulin is rather modest.

For comparison, the authors should include a discussion of how the ultra-long-acting insulin icodexin achieves its remarkable effect.

The authors are somewhat selective in their citation of prior literature on insulin total synthesis.

Reviewer #2 (Remarks to the Author):

This is an excellent paper with tremendous care observed during experiments. This is a hallmark of the

Iwaoka grp

The following issues need to be addressed

1) Correct the word "juice" in to homogenate

2) Provide a atomic/molecular reconstruction (diagram), if possible, of the mechanism by which the SSe substitution induced oligomerization

Reviewer #3 (Remarks to the Author):

Background

The authors sought to extend from their previous report where they demonstrated a successful synthesis of a novel seleno-insulin analogue with an increased resistance towards degradation by the insulin degradation enzyme (IDE). The sound multi-technical approach of the study culminated in the optimization conditions which favours the percentage yield of seleno-insulin. After optimization, the authors then thoroughly investigated the stability of resultant seleno-peptide, through looking at its resistance to insulin degrading enzyme, thermostability and oligomerization. From these studies, seleno-bridges offered an increased stability of the peptide in comparison to the BPI counterparts. Interestingly, the Sen-Ins was shown to also form oligomers with the native insulin, which could also mean this would lead to an extended lifespan of the native protein when these are co-administered. Extending further, the authors demonstrated its heightened lifespan in the human plasma which can support supports stability afforded perhaps by its seleno-bridges and oligomerization. Lastly, the authors sought to acutely investigate the pharmacodynamics of this analogue on a diabetic type 1 mimicking model. In general, the pharmacodynamics observation resembled those of a typic long-term insulin analogue.

Comments

In my opinion, the authors have conducted satisfactory experiments to justify some of the claims made in this paper. Authors have tried to prove beyond reasonable doubt the extended stability of the seleno-insulin analogues which is supported by evidence. The work presented seek to further promote innovation in the field of diabetes mellitus management, which I think is still relevant considering the upward trajectory of the disease prevalence coupled with its chronic nature. In my view, the work presented may offer a significant contribution in pharmaceutical sciences due to its innovative nature in improving the half-life of "biologics" or peptide-based therapeutics which are often administered through injections. In general, I found the paper well formulated. I would however request that the authors consider or address the following comments which are mainly on the pharmacodynamic aspects of the paper.

1. The authors investigated various concentrations of seleno-insulin on both diabetic and non-diabetic. In the graphs presented, there are no control groups, or sham treated groups. The lack of this group is of a concern for concrete judgements to be made. I would, therefore, recommend the inclusion of this group.
2. The authors utilized insulin lispro "short acting insulin" as their standard drug. Inclusion of a standard

drug was necessary; however, I do not think the use of a short acting alone sufficed in this case. I would perhaps suggest, the addition of a long-acting insulin: i.e glargine or detemir. In the absence long-acting insulin, interpretation of these observations maybe be exaggerated. Hence, having a long-acting insulin as a standard would assist in revealing the true robustness of the novel analogue on its extended glycaemic control. Perhaps the comparison would highlight extended advantage of the seleno-analogue provide over the conventional long-acting insulin counterparts.

3. Considering my previous comment, I do appreciate the inclusion of lispro for one reason. It has drawn my attention to the observation that seleno-insulin at almost all concentrations immediately decreased blood glucose concentration in a comparable manner to lispro. I am intrigued by this observation for various reasons. Firstly, considering the proposed oligomerization of insulin especially at higher concentrations, how would authors explain this observation, an immediate decrease in blood glucose concentration. Insulin oligomers, in general do not elicit a pharmacodynamic response. In the supplementary notes, the authors did investigate the stability of seleno-insulin in the blood, however I think confirming oligomerization in the blood is critical and would perhaps shed light on this concern I am raising. This further raises questions of whether oligomerization can be replicated in a blood setting, or whether the extended behavior on glucose lowering effect is mainly due to its IDE resistance properties. Furthermore, I believe the pharmacokinetic studies may provide more insights. I do take it into consideration that authors mention pharmacokinetics is in their future plans.

4. Statistical analysis: I would like to invite the authors to include a section on the manuscript detailing statistical protocols they followed.

Response to Comments

Manuscript ID: COMMSCHEM-23-0389A

Title: Diselenide-bond replacement of the external disulfide-bond of insulin increases its oligomerization that leads to sustained activity

Authors: Kenta Arai, Masaki Okumura, Young-Ho Lee, Hidekazu Katayama, Kenji Mizutani, Yuxi Lin, Sam-Yong Park, Kaichiro Sawada, Masao Toyoda, Hironobu Hojo, Kenji Inaba, Michio Iwaoka

Reviewers' comments:

Reviewer #1

The present study builds on previous reports by the group on the synthesis and characterization of seleno-insulin and other insulins. The paper reports an improved synthesis of the original seleno-insulin variant, where the main part is the optimized two-chain oxidative folding.

It is an interesting study and performed at a high level.

Response:

We gratefully acknowledge the careful reading and pertinent comments of the reviewer for our manuscript. We have addressed the reviewer's comments, accordingly, as described below.

(1) The abstract indicates that seleno-insulin is a single insulin variant, while the main text mentions the other seleno-insulins. It would be good if the authors could be more clear about that seleno-insulins is a class of compounds (insulins), where the authors reported the first of these.

Response:

We thank the reviewer for this valuable suggestion. The abstract in the current manuscript has been modified to state that seleno-insulin is a class of artificial insulin variants and that **SeIns** (i.e., a [C7U^A,C7U^B] analog of BPIs), which is focused on this paper, is the first example of the seleno-insulins that we have pioneered. <page 2, lines 40-43>

(2) It also includes an analytical ultracentrifugation study on the oligomerization the synthesized seleno-insulin. The authors hypothesize that they can observe the effect of formation of hetero-oligomers of insulins. However, while this may be correct, they do not have direct evidence for this.

Response:

As this reviewer pointed out, unfortunately, direct observation of the hetero-oligomer formation of insulin analogs has not been achieved at present. In the meantime, we have already verified by cryo-EM as

described in Ref. 16 that wild-type insulin without SeSe bond can essentially form a dimer or hexamer even in the absence of zinc ions (Fig. R1). Since the monomer structures of Selns and BPIns are very similar to each other [Ref. 9], formation of hetero-oligomer between Selns and BPIns would be possible. Nevertheless, we are currently trying to crystallize the hetero-oligomers for X-ray crystallography to obtain the direct evidence and to investigate the molecular mechanism of oligomer formation.

Fig. R1: Cryo-EM analysis of wild-type human insulin. (a) Crystal structure of a insulin hexamer, T₆ state, with zinc ions (PDB: 1MSO). (b) Representative 2D class averages of single particle cryo-EM analysis of wild-type insulin without zinc ion. Image processing visualized converged 2D class-averaged images, showing stable structures with a diameter of about 50 Å, which corresponds to a hexameric complex similar to T₆ oligomer (a). Data were quoted from Ref. 16.

(3) It also includes a careful study of resistance to processing by IDE, in response to a paper by Weiss et al, which contradicted their original results. The results reported here seem trustworthy. Could the authors more explicitly discuss why their findings differ from Weiss et al.?

Response:

We recognize that this is an important question. However, we are unfortunately unable to provide a definitive answer at present to this question of why the results showed by Weiss *et al.* are completely different from our results. Therefore, we would like to refrain from address this point in the manuscript. One seeming possibility is the difference in the concentrations of Selns as discussed in page 7, lines 162-164 in the revised manuscript.

(4) The authors comment that their seleno-insulin likely has low short-term toxicity. However, there seems to be no comment on the possible toxicity with longer-term use. This is an important question, as T1D is chronic and requires daily injections. The authors really should comment on this.

Response:

To address this reviewer's concern, we have added descriptions as to importance and needfulness of further investigation on the longer-term administration of Selns. See page 15, lines 336–338 in the revised manuscript.

(5) The authors state that this work could contribute to the development of longer-acting insulins. While the results on the inhibitory effect on IDE (insulin degrading enzyme), which already was part of the original publication, is interesting and very relevant, there is a lot more to making a long-acting insulin.

Response:

We thank the reviewer for this encouraging comment. We have added descriptions related to the needfulness of further accumulation of basic data for practical use of Selns. See the yellow-highlighted part in the conclusion section (page 18, lines 404-405 of the revised manuscript).

(6) The seleno-insulin was assembled by chemical synthesis, whereas the approved long-acting insulins are all made by expression. Chemical synthesis would be more costly, as multi-ton amounts are required. The author should consider this.

Response:

To address the reviewers' concern, we have described the advantages and disadvantages of chemical synthesis in terms of its industrial and commercial potential. Please see the yellow-highlighted part in the conclusion section (page 17, lines 376–381 of the revised manuscript).

(7) Also, the long-acting effect achieved with their seleno-insulin is rather modest. For comparison, the authors should include a discussion of how the ultra-long-acting insulin icodex achieves its remarkable effect.

Response:

We gratefully acknowledge the valuable comment and suggestions of the reviewer to improve the quality of our manuscript. We have now clearly described the differences in the molecular mechanisms by which these insulin analogs exert their long-acting effects. Please see the yellow-highlighted part in the conclusion section (page 17, lines 396–400 of the revised manuscript).

(8) The authors are somewhat selective in their citation of prior literature on insulin total synthesis.

Response:

We have cited several additional original papers that are important in the history of the development of the chemical synthesis of insulin. Please see references 21, 22, 24, 25, 26, and 29 in the revised manuscript.

Reviewer #2 (Remarks to the Author):

This is an excellent paper with tremendous care observed during experiments. This is a hallmark of the Iwaoka grp.

Response:

We thank the reviewer for their high evaluation for our manuscript.

The following issues need to be addressed

1) Correct the word "juice" in to homogenate

Response:

According to your suggestion, the word "juice" was replaced by "homogenate". Please see page 3, lines 64

in the revised manuscript.

2) Provide a atomic/molecular reconstruction (diagram), if possible, of the mechanism by which the SSe substitution induced oligomerization

Response:

As we mentioned in our response to reviewer 1's second comment, we are currently investigating on X-ray crystallographic analysis of homo- and hetero-oligomers of seleno-insulin analogs. We already have some inferences about the mechanism of oligomerization based on the experimental results, but it needs more works to obtain clear evidence. The details will be reported in our next papers.

Reviewer #3 (Remarks to the Author):

Background

The authors sought to extend from their previous report where they demonstrated a successful synthesis of a novel seleno-insulin analogue with an increased resistance towards degradation by the insulin degradation enzyme (IDE). The sound multi-technical approach of the study culminated in the optimization conditions which favours the percentage yield of seleno-insulin. After optimization, the authors then thoroughly investigated the stability of resultant seleno-peptide, through looking at its resistance to insulin degrading enzyme, thermostability and oligomerization. From these studies, seleno-bridges offered an increased stability of the peptide in comparison to the BPI counterparts. Interestingly, the Sen-Ins was shown to also form oligomers with the native insulin, which could also mean this would lead to an extended lifespan of the native protein when these are co-administered. Extending further, the authors demonstrated its heightened lifespan in the human plasma which can support supports stability afforded perhaps by its seleno-bridges and oligomerization. Lastly, the authors sought to acutely investigate the pharmacodynamics of this analogue on a diabetic type 1 mimicking model. In general, the pharmacodynamics observation resembled those of a typical long-term insulin analogue.

Comments

In my opinion, the authors have conducted satisfactory experiments to justify some of the claims made in this paper. Authors have tried to prove beyond reasonable doubt the extended stability of the seleno-insulin analogues which is supported by evidence. The work presented seek to further promote innovation in the field of diabetes mellitus management, which I think is still relevant considering the upward trajectory of the disease prevalence coupled with its chronic nature. In my view, the work presented may offer a significant contribution in pharmaceutical sciences due to its innovative nature in improving the half-life of “biologics” or peptide-based therapeutics which are often administered through injections. In general, I found the paper well formulated. I would however request that the authors consider or address the following comments which are mainly on the pharmacodynamic aspects of the paper.

Response:

We thank the reviewer for their valuable comments and suggestions for our manuscript. We considered or

addressed the reviewer's requests accordingly, as described below.

1. The authors investigated various concentrations of seleno-insulin on both diabetic and non-diabetic. In the graphs presented, there are no control groups, or sham treated groups. The lack of this group is of a concern for concrete judgements to be made. I would, therefore, recommend the inclusion of this group.

Response:

In the current manuscript, the blood glucose changes observed in normal and diabetic rats when diluent (saline) was administered have been incorporated into Fig. 5 as control data.

2. The authors utilized insulin lispro "short acting insulin" as their standard drug. Inclusion of a standard drug was necessary; however, I do not think the use of a short acting alone sufficed in this case. I would perhaps suggest, the addition of a long-acting insulin: i.e glargine or detemir. In the absence long-acting insulin, interpretation of these observations may be exaggerated. Hence, having a long-acting insulin as a standard would assist in revealing the true robustness of the novel analogue on its extended glycaemic control. Perhaps the comparison would highlight extended advantage of the seleno-analogue provide over the conventional long-acting insulin counterparts.

Response:

We thank the reviewer for their insightful comment. We are also very interested in comparing the pharmacological effects of seleno-insulin with those of commercially available long-acting insulins, such as glargine and detemir. Unfortunately, however, due to several matters in animal experiments at this time, it is currently difficult to immediately conduct experiments with the formulations and add the results in the text.

The primary purpose of this experiment is to verify whether S to Se substitution enhances the intrinsic sustained effect of the insulin molecule, and, most importantly, to compare the hypoglycemic behavior between wild-type and seleno-insulin. Nevertheless, we are currently designing and planning new animal studies and will conduct pharmacodynamic and pharmacokinetic analyses along with several new seleno-insulin analogs and will report those results in the next papers.

3. Considering my previous comment, I do appreciate the inclusion of lispro for one reason. It has drawn my attention to the observation that seleno-insulin at almost all concentrations immediately decreased blood glucose concentration in a comparable manner to lispro. I am intrigued by this observation for various reasons. Firstly, considering the proposed oligomerization of insulin especially at higher concentrations, how would authors explain this observation, an immediate decrease in blood glucose concentration. Insulin oligomers, in general do not elicit a pharmacodynamic response. In the supplementary notes, the authors did investigate the stability of seleno-insulin in the blood, however I think confirming oligomerization in the blood is critical and would perhaps shed light on this concern I am raising. This further raises questions of whether oligomerization can be replicated in a blood setting, or whether the extended behavior on glucose lowering effect is mainly due to its IDE resistance properties. Furthermore, I believe the pharmacokinetic studies may provide more insights. I do take it into

consideration that authors mention pharmacokinetics is in their future plans.

Response:

We would like to express our sincere gratitude to the reviewer for guiding our research. Indeed, it is interesting to us that an immediate hypoglycemic effect was observed. Generally, commercial long-acting insulins form soluble or insoluble homo-/hetero-oligomers subcutaneously or in the blood, and hence delays release of the monomers in the blood stream, thereby decreasing blood glucose levels in a gentle and sustained manner. On the other hand, our seleno-insulin rapidly decreased blood glucose levels at a rate comparable to wild-type and a rapid-acting insulin, and also exhibited sustained pharmacological effects. This would suggest that the mechanism of the sustained efficacy is different from those of conventional long-acting insulins and may imply that the sustained behavior of pharmacological effects is mainly due to IDE resistance. However, as already indicated in our response to comment 3, this inference cannot be immediately verified by animal studies. As this reviewer mentioned, we also recognize that the pharmacokinetic analyses of seleno-insulin and commercial insulins is an important issue that should be given the highest priority in our future research. See page 16, lines 369-371.

4. Statistical analysis: I would like to invite the authors to include a section on the manuscript detailing statistical protocols they followed.

Response:

According to the reviewer's suggestion, statistical protocols for animal experiments have been added to the current experimental section. Please see section "Statistical analyses" in Methods (see page 20, lines 487-489).

List of Changes

Abstract

1. According to reviewer 1's suggestion, minor corrections have been made to the abstract in the revised manuscript. See the text highlighted in yellow.

Introduction

2. "juice" → "homogenate" <page 3, line 64>

Results and Discussion

3. Control data for blood glucose changes in rats have been incorporated into Fig. 5. <page 14, Fig.5>
4. With the addition of the control data in Fig. 5, the caption for Fig.5 has also been slightly changed <page 14, Fig.5>.
5. The sentence "Needless to say, it is also important to investigate long-term effects of Selns in future study because daily administration of insulin is required for diabetic patients." has been inserted into the end of the first paragraph of the section "**Selns exerts a sustained hypoglycemic effect.**" <page 15, lines 336-338>

Conclusions

6. The sentences “Considering the application to industrial production, the chemical synthesis of insulin would be more costly than the current protein expression method using the recombinant DNA. However, using the chemical method, such as the double-chain assembly optimized in this study, introduction of site-specific mutations with natural and non-natural amino acid residues should be easier, hence it would help speed up the structural screening for development of insulin formulation candidates. On the other hand,” has been inserted in the first paragraph. <page 17, lines 376-381>
7. The sentences “In addition, insulin icodec, a once-weekly formulation, which is conjugated with C20 fatty diacid (icosanedioic acid) can exhibit an ultra-long acting property due to its extremely strong association with albumin, and is undergoing clinical trials for use for new basal insulin therapy^{57–59}. Although the sustained potency of our SeIns would be mild compared to this insulin analog,” has been inserted in the second paragraph. <page 17, lines 396-400>
8. The sentence “Thus, development of SeIns analogs is a promising strategy for the design of new long-acting insulins.” in the original draft has been changed to “Thus, development of SeIns analogs is a promising strategy for the design of new long-acting insulins, although additional basic data, such as long-term toxicity tests, need to be accumulated in preclinical studies.” in the current draft. <page 18, lines 404-405>

Methods

9. Section for Statistical analyses has been added in the current draft. <page 20, lines 487-489>

References

10. New references (refs. 21, 22, 24, 25, 26, 29, 57, 58, and 59) have been added to the revised manuscript, and the references 21-62 were renumbered accordingly.

All revisions made to the resubmitted manuscript are highlighted in yellow.

REVIEWERS' COMMENTS:

Reviewer #1 (Remarks to the Author):

This referee appreciates the revisions that the authors have made and now recommends publication.

Reviewer #3 (Remarks to the Author):

The authors have carefully addressed comments or suggestions made to my satisfaction